# Catheter Ablation for Atrial Fibrillation in Structural Heart Disease: A Review

**DOI:** 10.3390/jcm12041431

**Published:** 2023-02-10

**Authors:** Francesco Maria Angelo Brasca, Roberto Menè, Giovanni Battista Perego

**Affiliations:** 1Department of Cardiovascular Neural and Metabolic Sciences, San Luca Hospital, IRCCS Istituto Auxologico Italiano, 20149 Milan, Italy; 2Department of Medicine and Surgery, University of Milano-Bicocca, 20126 Milan, Italy

**Keywords:** atrial fibrillation, catheter ablation, structural heart disease

## Abstract

Atrial fibrillation (AF) is the most common arrhythmia encountered in clinical practice. Patients with structural heart disease (SHD) are at an increased risk of developing this arrhythmia and are particularly susceptible to the deleterious hemodynamic effects it carries. In the last two decades, catheter ablation (CA) has emerged as a valuable strategy for rhythm control and is currently part of the standard care for symptomatic relief in patients with AF. Growing evidence suggests that CA of AF may have potential benefits that extend beyond symptoms. In this review, we summarize the current knowledge of this intervention on SHD patients.

## 1. Introduction

Atrial fibrillation (AF) is the most prevalent arrhythmia in clinical practice [1] and it is even more frequent in patients with structural heart disease (SHD) [2]. 

SHD encompasses a heterogeneous group of patients that share some important features, whatever the underlying disease. First, they are characterized by reduced hemodynamic tolerance to elevated heart rates and/or to the loss of atrial contribution to LV filling, which are associated with AF; second, in this population, the choice of anti-arrhythmic drugs (AADs) is limited, owing to possible side-effects; and third, the probability of rhythm control success on AADs is lower than in non-SHD patients [3]. Additionally, SHD patients are less represented in large clinical trials on AF catheter ablation (CA); therefore, except for a suggested higher recurrence rate [4], little evidence is available concerning interventional management of these patients.

The purpose of this article is to briefly review the clinical knowledge on interventional treatment of AF in this population.

### Structural Heart Disease Definition

“Structural heart disease” (SHD) is an over-reaching term first introduced by Martin Leon at the 1999 Transcatheter Cardiovascular Therapeutics Meeting to encompass all cardiac disease processes [5].

The European Society of Cardiology (ESC) guidelines identify AF as secondary to SHD when a left ventricle (LV) systolic or diastolic dysfunction is demonstrated or LV hypertrophy, valvular disease, and/or other SHDs are documented [6]. Subsequent literature evolved the nomenclature so that, currently, SHD includes: (a) heart failure with reduced ejection fraction (HFrEF, previously severe or moderate LV systolic dysfunction); (b) heart failure with preserved ejection fraction (HFpEF, previously LV diastolic dysfunction); (c) valvular heart disease (VHD), ranging from prosthetic valves to rheumatic ones; and (d) specific cardiomyopathies, such as hypertrophic cardiomyopathy (HCM) [7].

According to the ESC Guidelines definition, criteria to identify SHD patients are based on non-specific parameters [7]. Thus, the SHD impact on AF prevalence and patient’s prognosis could be different according to the underlying pathology and the severity of the disease.

In studies dealing with AF, specific biomarkers to assess hemodynamic status (e.g., natriuretic peptides) are rarely available and seldom reported and the degree of atrial remodeling is not uniformly defined because of the use of different parameters and imaging techniques (e.g., echocardiography or magnetic resonance).

## 2. Impact of SHD on AF

Independently of the underlying disease, a common final pathway is supposed to lead to AF: elevated left atrial pressure, causing “atrial myopathy” [8]. Indeed, atrial hypertension is associated with chamber dilation, extracellular matrix remodeling, autonomic imbalance, and calcium handling defects, which have a well demonstrated proarrhythmic effect and are involved in the induction and maintenance of AF [9,10].

The actual incidence and prevalence of AF in the overall SHD population have not been assessed, but data are available for specific subgroups.

### 2.1. HFrEF

In 2003, Maisel estimated that the AF prevalence ranges from <10% in New York Heart Association (NYHA) functional class I patients to nearly 50% in NYHA IV patients. Overall, HF patients have a sixfold increased risk of AF in the long term [11].

Both AF and HF have a higher prevalence in the elderly, and this might partially explain the correlation between the degree of functional impairment and the occurrence of AF. Nevertheless, there are several pathophysiological reasons for why they are supposed to favor each other, leading to the concept that “AF begets HF and vice versa” [12].

In particular, myocardial inflammation and fibrosis, leading to atrial interstitial fibrosis, are present in both AF and HF. Thus, during exertion, AF itself could be less tolerated in HF patients and, thereby, may trigger clinical recognition of this condition.

In a sample from the Framingham cohort including data between 1980 and 2012, it was found that a greater proportion of individuals have AF without HF, and AF more commonly precedes HF than in cohorts studied in previous years [12]. However, different strategies to detect AF have been implemented over time, making these results hardly comparable. Regarding the temporal relationship between AF and HF onset, it has been noted that patients who develop HF first and AF later have a worse clinical progression compared with the opposite [13].

### 2.2. HFpEF

HFpEF often coexists with other cardiac diseases; thus, it might be difficult to isolate its specific effect. Diastolic dysfunction should be graded according to American Society of Echocardiography recommendations and the evaluation should be based on parameters that are not affected by the presence of AF [14].

In a population study by Chen et al., one-third of patients with isolated diastolic dysfunction and HF-related symptoms show AF in the ECG presentation [15].

In a longitudinal study by Tsang and colleagues, abnormal LV diastolic function is associated with new onset of AF in almost 10% of cases within 4 years. In particular, the presence of grade 2 or 3 diastolic dysfunction was associated with a 2.5-fold increase in AF recurrence risk when compared with grade 1 dysfunction or normal diastolic function [16].

### 2.3. Valvular Heart Disease (VHD)

The presence of rheumatic mitral stenosis, repaired mitral valve or prosthetic valve are three different conditions. The presence of just one of the three is sufficient to distinguish valvular and non-valvular AF [17].

In 1990, data from surgical series were published by Wipf et al., who reported an AF prevalence close to 75% in rheumatic heart disease (RHD) at the time of surgical treatment [18]. In addition, in early studies on patients treated with AAD, AF frequently complicated RHD, with more than 30% of patients with AF episodes over long-term follow-up [19].

Even if a decline in RHD prevalence was recorded in Western countries, the presence of a valvular heart disease is still associated with a 1.8–3.4-fold increased risk for AF [20], and mitral stenosis and mechanical prosthetic valves are associated with a further increase in thromboembolic risk [21].

### 2.4. Cardiomyopathies

Hypertrophic cardiomyopathy (HCM) is the most common hereditary cardiomyopathy [22], and thus the best investigated. The estimated prevalence of AF in HCM is 22.5% and the annual incidence is 3.1% [23].

The maintenance of sinus rhythm could be of a particular importance in HCM patients, as this is associated with a significant improvement in the New York Heart Association functional (NYHA) class and quality-of-life score [24]. These benefits seem to depend on heart rate control and atrial active contraction, which increase the LV filling, reducing outflow obstruction.

Furthermore, dyspnea and other heart failure symptoms are frequently associated with AF, which is a major cause of hemodynamic deterioration and an ominous prognostic indicator [24].

## 3. Catheter Ablation

Most structural heart diseases are associated with some degree of atrial hypertension, remodeling, and fibrosis. Marrouche and colleagues have reported that extensive atrial fibrosis is associated with a significant decrease in AF ablation effectiveness [25]. Thus, patients with SHD are expected to have a high rate of AF recurrence and to be less responsive to CA compared with non-SHD patients [26].

The evaluation and comparison among studies on CA in SHD is made difficult by the differences in the ablation strategy, energy sources, and endpoints.

To date, pulmonary veins’ isolation (PVI) is recommended for the index procedure in both paroxysmal AF (PAF) and persistent AF (PerAF) patients [27]. More extensive ablation strategies are not supported by consistent data [28]. In particular, no evidence supports a different approach in SHD patients, even if, in this population, there is a widespread tendency to extend ablation beyond PVI, adding lines or complex fragmented atrial electrogram (CFAE) and extra-pulmonary foci ablation.

Radiofrequency is the most represented energy source in the literature on SHDs, whereas cryoballoon results have been reported only in registries [29].

Finally, studies are hardly comparable because of the differences in the assessment of AF recurrence. In many cases, AF detection is based on symptoms or on electrocardiography and/or prolonged cardiac monitoring triggered by symptoms. In some instances, loop recorders are implanted, providing continuous monitoring throughout the follow-up. In the specific setting of SHD, AF is more often symptomatic, thus symptom-based detection might be somehow more reliable than in the general population.

### 3.1. CA in HFrEF

AF ablation in HFrEF patients has been studied in several randomized clinical trials (RCTs). CA was compared either to other rhythm control strategies (AADs) or to rate control (drugs and/or ablate and pace). In most studies, extrapulmonary lesions, such as left atrial lines or CFAE, were added to PVI; single or multiple procedures were possible. The endpoints ranged from symptom recurrence, documented AF recurrence, and functional improvement (NYHA class, 6 min walk distance, QoL scores, peak oxygen consumption, LVEF, and BNP) to hard ones such as mortality and hospitalization.

Table 1 shows the results of the main studies, with some of them deserving specific considerations.

The CAMERA MRI trial included only idiopathic systolic dysfunction and used cardiac magnetic resonance to assess LVEF [30]. A significant improvement in LVEF (18 ± 13%) in the group of patients treated with catheter ablation and a reduction in LV end-systolic volume (−24 ± 24 mL/m^2^ vs. −8 ± 20 mL/m^2^, *p* < 0.0001) were observed; furthermore, extensive late gadolinium enhancement (LGE) was a negative predictor of LV functional improvement. Of note, MacDonald et al. published a similar study that failed to show an improvement in LVEF and other functional endpoints, probably owing to a larger proportion of advanced HF patients (90% NYHA III or more) [31].

More recently, RCTs comparing CA to AADs for rhythm control strategy focused on hospitalization or mortality. The AATAC trial compared CA to amiodarone in patients with PerAF [32]. An implanted device was used to detect AF. CA was more effective in preventing recurrences (30% vs. 66%; *p* < 0.001), preventing unplanned HF hospitalization (31% vs. 57%; *p* < 0.001), and reducing all-cause mortality (8% vs. 18%; *p* = 0.037).

CASTLE-AF enrolled 363 HFrEF patients, randomized to PVI ablation or medical (both rate and rhythm control) therapy [33]. Patients had PAF or PerAF, LVEF < 35%, NYHA functional class equal to II or greater, an ICD or CRTD device, and should have failed a prior treatment with AAD. At 38-month follow-up, CA reduced the risk of death or HF hospitalization (28.5% vs. 44.6%, *p* = 0.006). Both all-cause mortality (13.4% vs. 25%, *p* = 0.01) and cardiovascular death (HR 0.49, *p* = 0.009) were significantly reduced in the ablation arm. Arrhythmia-free survival at 5 years was 63% in the ablation group and 22% in the medical therapy arm. Nevertheless, only 10% of screened patients were included in the study, so these results could be applied only in really selected patients.

Interestingly, in a sub-analysis of CASTLE-AF [34], Brachman and colleagues found that a reduction in AF burden below 50% after 6 months of catheter ablation was associated with a significant reduction in all-cause mortality and hospitalizations for HF. The same relationship could not be found if patients were stratified according to AF recurrence after ablation, defined by the HRS consensus statement of at least one AF episode longer than 30 s following the procedure [4]. The authors speculate that this might be explained by a survival benefit proportionate to the time spent in sinus rhythm and by the reduction in AF burden, being an epiphenomenon of reverse atrial and ventricular remodeling following ablation. Overall, these considerations may prompt a paradigm shift in how procedural efficacy is defined.

In 2019, the AMICA trial was stopped owing to futility because a similar improvement in LVEF was obtained in both the CA group and the medical therapy group [35]. Of note, LVEF improved more than expected from the literature in the control group.

Lastly, the RAFT-AF trial compared all-cause mortality and HF events in both HFrEF and HFpEF patients with AF randomized to ablation-based rhythm control or to pharmacologic rate control [36]. Despite showing a non-significant trend for improved outcomes with ablation-based rhythm control (29% relative risk reduction, *p* = 0.066), the trial was stopped early for apparent futility. It must be noted that the decision to terminate the trial was taken following the 2017 ad-interim analysis, when the available results found a trend for a worse outcome with CA, which was eventually overturned in the final results. Notwithstanding this, ablation-based therapy was associated with significantly greater gains in quality of life, 6 min walk distance, and LVEF. In addition, there was a significantly greater fall in NT-proBNP levels in the ablation group.

The results of these trials have been included in a 2020 meta-analysis that, in a population of 1112 HF patients, has demonstrated a consistent benefit of CA compared with AADs in terms of all-cause mortality (49% relative risk reduction (RRR), *p* = 0.0003), re-hospitalizations (56% RRR, *p* = 0.003), LVEF improvement (mean improvement of 6.8%, *p* = 0.0004), AF/AT recurrence (96% RRR, *p* < 0.00001), and quality of life (*p* = 0.007), without significant differences concerning safety [37]. Overall, these results highlight the physiologic and clinical advantage of maintaining sinus rhythm in HF patients, as well as the effectiveness of CA in pursuing this objective. Interestingly, in the same meta-analysis, a second subset of studies comparing pharmacologic rhythm control to rate control failed to demonstrate clinically significant benefits. This may be explained by a lower efficacy of AADs in maintaining sinus rhythm and by the neutralization of the benefits of sinus rhythm by the adverse effects of these medications.

Furthermore, any rhythm control strategy is at high risk of failure when used in too advanced stages of the disease [38].

### 3.2. CA in HFpEF

Few data are available on CA in patients with HFpEF because of the recent definition of this nosologic entity.

In the study by Cha et al., the 1-year arrhythmia-free survival after CA was 84% in patients with normal LV function and significantly lower when diastolic or systolic dysfunction was found at echocardiography (75% and 62%, respectively) [39]. Of note, in the diastolic dysfunction group, patients were older and more frequently had hypertension. Nevertheless, both systolic and diastolic dysfunction were significant predictors of increased AF recurrence risk, even after correction for these potential confounders.

Hu et al. showed an association between diastolic abnormality, low voltages at LA electro-anatomical map, and recurrence rates [40]. Even if limited by the low number of patients enrolled, this study suggests a pathophysiological link between extensive fibrosis and CA failure in HFpEF.

A meta-analysis on six observational studies comparing CA in HFpEF and HFrEF found no differences in terms of procedural efficacy, periprocedural adverse events, or re-hospitalizations between the groups, but highlighted a significantly lower mortality at follow-up in the HFpEF group (mean difference of 0.41; 95% CI 0.18–0.94) [41].

Using retrospective data from a national administrative database, Krishnamurthy and colleagues found that CA in patients with HFpEF, compared with patients without HF, is associated with more procedural complications, all-cause readmissions, cardiac readmissions, noncardiac readmissions, and early mortality. Nevertheless, when adjusting for age, sex, and comorbidities, only all-cause readmissions maintain a statistically significant increased risk (OR 1.52; *p* = 0.002) [42]. These results suggest that increased procedural complications, readmission, and early mortality following CA in HFpEF patients are mainly driven by concomitant risk factors, such as age and comorbidities, rather than by HFpEF itself.

Finally, a significant group of HF patients was included in the CABANA trial and randomized to catheter ablation versus drug therapy [43]. In the related sub-group analysis of 778 patients with HF [44], 91% of these had LVEF > 40% and 79% had LVEF > 50%. This sub-group analysis can thus be considered the first randomized prospective collection of data on AF ablation in HFpEF. A significant reduction in the composite outcome of death, disabling stroke, serious bleeding, and cardiac arrest (HR: 0.64, 95% CI 0.41–0.99), as well as in all-cause mortality alone (HR: 0.57, 95% CI: 0.33−0.96), was found; notably, these beneficial results were not evident in the main trial including both HF and non-HF patients. In addition, patients undergoing CA experienced a considerable improvement in quality of life indicators and, not surprisingly, a lower incidence of AF recurrence and burden in each of the 12-month follow-up assessments to the end of the 5-year observation period. Interestingly, compared with other studies, patients were randomized to treatment within a relatively short period of time from their diagnosis of AF (median 1.1 years); as it is known that rhythm control pursued early in the course of AF is associated with better outcomes and that CA is more effective than AADs in maintaining sinus rhythm [45], part of the beneficial effects seen in the CABANA subgroup may be explained by early intervention. Overall, these data reinforce the role of CA in HFpEF, especially when administered in the early stages of the disease.

### 3.3. Valvular Heart Disease (VHD)

Few studies on CA in patients with uncorrected VHD are available, owing to the strong indication of valvular defect correction before trying the invasive treatment of AF [46]. The results are not consistent; a study found no difference in arrhythmias recurrence between VHD patients and non-VHD patients, but, notably, the recurrence rate was higher in patients with larger left atria in both groups [47]. Nevertheless, in moderate VHD patients, AF recurrence is more frequent than in non-VHD patients after discontinuation of AADs in the long-term follow-up [48].

Surgical ablation during valve surgery is a valid option [49] and its results are superior to a subsequent single CA procedure [50], but this analysis extends beyond the aim of this paper.

Catheter ablation in patients with prosthetic valves remains challenging; lower effectiveness, higher complication rates, greater radiation exposure, and higher incidence of post-ablation atrial tachycardia were reported [50,51]. Furthermore, a possible role of non-PV foci was suggested by the evidence that a strategy including extended PVI and non-PV trigger elimination is associated with a higher 12-month and long-term arrhythmia-free survival [52].

According to a meta-analysis by Santangeli et al., CA of valvular AF is associated with an increased risk of recurrences in patients with MVR, but it is feasible and safe, despite the presence of prosthetic valves or annuloplasty rings, in experienced centers [53].

### 3.4. Hypertrophic Cardiomyopathy (HCM)

To date, the role of AF ablation in this setting needs to be inferred from non-randomized observational studies, because no RCT is available. The success rate of AF ablation is lower than in patients without HCM [54]. The procedural results were evaluated in a meta-analysis by Zhao et al. [55]; a single ablation procedure is frequently followed by arrhythmia relapses and antiarrhythmic drugs and/or multiple procedures could be required. In particular, the probability of 3 months’ freedom from arrhythmias after a single procedure is estimated to be 79%, while at 18 months, most patients experienced AF recurrences.

The association between AF recurrence and wall thickness or left ventricle (LV) outflow tract obstruction is not predictive, while left atrial (LA) structure, diameter, and electrical features, as well as the presence of LV apical aneurysm, seem to predict post procedural outcome [55,56,57].

Besides the possible presence of gaps in the isolation lesions set, two interesting hypotheses are proposed to explain the higher recurrence rate seen in HCM patients: the response of hypertrophic tissue to RF is different from that of normal myocardial tissue and non-PV foci are more frequent in HCM patients. The former might be the cause of PV stenosis, which occurs more often in these patients [58]. The latter is supported by the high frequency of post-ablation non-AF atrial tachycardias (38.4%) that have either macro-reentry or localized reentry as an underlying mechanism [59]. This may be explained by the extensive structural alterations seen in HCM atria and may be a reason to pursue a non-PVI-only ablation strategy [60].

### 3.5. CA Technique in SHD

Following the description of pulmonary veins’ isolation (PVI) as the first effective CA strategy for AF [61], different techniques have been described in terms of both energy source (e.g., radiofrequency, cryo-energy, and electroporation) and ablation targets beyond PVI (e.g., posterior wall isolation and rotor ablation). Despite a flourishing body of literature on the role of these different techniques in AF, all of the aforementioned trials have been carried out using radiofrequency as ablation energy and few studies have specifically addressed the topic of CA strategies in SHD.

Recently, a retrospective analysis of the ONE-Stop Italian registry on Cryoballoon (CB) PVI was published [29]. The procedure time, fluoroscopic time, and complication rate were not different in a subgroup of 282 SHD patients as compared with the non-SHD cohort. The recurrence rate was similar in both groups at 13-month follow-up (22.0% vs. 21.6%; *p* = 0.895) and was not related to either left atrial size or LVEF. Of interest, the percentage of SHD patients on AAD treatment decreased from 70.7% to 28.7% after CB-PVI (*p* = 0.001). Because of its retrospective nature, the study suffers from possible selection bias and included SHD patients with minimal or no reduction in LVEF. Similar results have been found in an international cohort including 318 patients with HF; that is, procedure-related safety and long-term efficacy following PVI through cryoablation were comparable in patients with and without HF [62]. Altogether, these results suggest that CB-PVI is feasible and safe in the SHD setting and that additional benefits might be obtained through the reduction in AAD usage.

More extensive atrial remodeling is present in persistent AF (as opposed to paroxysmal) and when AF is associated with HF. Therefore, mechanisms other than PVs’ firing have been hypothesized for AF initiation and maintenance in these settings. Although meta-analyses suggest a potential incremental benefit in terms of procedural efficacy for extra-PVI lesions in patients with persistent AF [63,64], no conclusive evidence has been provided to date. Furthermore, in 2014, a meta-regression analysis reported no differences in sinus rhythm maintenance between the PVI-only approach and extended left atrial ablation in patients with AF and HFrEF [65]. Interestingly, however, in a small single-center study on paroxysmal AF, non-PV triggers were found to be more frequent in patients with LVEF < 35% compared with those with LVEF > 50% (69.1% vs. 26.6%; *p* < 0.001); ablation of these triggers, in addition to PVI, resulted in improved long-term procedural success (75.0% vs. 32.2%; *p* < 0.001) [66]. Thus, further investigation is needed to definitively assess the role of additional lesions in addition to PVI in these contexts.

## 4. Conclusions

Catheter ablation in SHD patients could be technically challenging, but it is feasible and safe. The best ablation strategy is not well defined; although non-PV additional lesions are a common practice, a PVI-only approach might have a role even in SHD patients. Whatever the initial ablation method, multiple procedures are often needed. Evidence of the improvement in LV function and quality-of-life is available in particular for HFrEF patients, while data on hospitalization and mortality are encouraging but limited to very specific subsets. Overall, despite that, in SHD patients, CA shows a higher AF recurrence rate, the clinical benefit could be more significant in the setting of SHD.

## Figures and Tables

**Table 1 jcm-12-01431-t001:** Main trials comparing CA for AF in HF. Bold character is for study primary outcome. Asterisks (*) denote statistical significance. AAD: anti-arrhythmic drugs; AFEQT: atrial fibrillation effect on quality of life; AVN: atrio-ventricular node; CRT: cardiac resynchronization therapy; EF: ejection fraction; ICD: implantable cardioverter-defibrillator; MLHFQ: Minnesota living with heart failure questionnaire; NYHA: New York Heart Association; PAF: paroxysmal atrial fibrillation; PerAF: persistent atrial fibrillation.

	PABA-CHF [67] (2008)	ARC-HF [68] (2013)	CAMTAF [69] (2014)	CAMERA-MRI [30] (2017)	AATAC [32] (2016)	CASTLE-AF [33] (2018)	CABANA Subgroup [44] (2021)	RAFT-AF [36] (2022)
Sample size	81	52	50	68	203	363	778	411
Population	EF < 40% NYHA II-III PAF/PerAF	EF < 35% NYHA II-IV PerAF	EF < 50% NYHA II-IV PerAF	EF < 45% (MR) NYHA II–IV PerAF No CAD	EF < 40% NYHA II-III PerAF ICD/CRT-D	EF < 35% NYHA II-IV PAF/PerAF ICD/CRT-D	NYHA II-IV PAF/PerAF >65 years or <65 years + 1 risk factor for stroke	NYHA II/III PAF/PerAF High NT-proBNP
Control group	AVN ablation + CRT	Rate control	Rate control	Rate control	Amiodarone	Pharmacologic rate or rhythm control	Pharmacologic rate or rhythm control	Pharmacologic rate control
Follow-up (months)	6	12	12	6	24	60	60	24
AF-free survival	71% AAD off	68% AAD off	38% AAD off	56% AAD off	**70% AAD off vs. 34% with AAD ***	63.1% AAD off vs. 21.7% with AAD *	73% AAD off vs. 42% with AAD *	85.6% AAD off vs. 12.9% with AAD *
LVEF	**35% vs. 28% ***	+5.6% *	**40% vs. 31% ***	**Δ 18% vs. 4.4% ***	Δ 8.1% vs. 6.2% *	Δ 8.0% vs. 0.2% *		10.1% vs. 3.8% *
Peak O_2_		**Δ +3.07 mL/kg/min ***	22 vs. 18 mL/kg/′ *					
BNP		Δ −124 vs. −18 pg/mL *	126 vs. 327 pg/mL *	98 vs. 247 pg/mL *				−77.1% vs. −39.2% *
6′ walk (m)	**340 vs. 297 ***	+21 vs. −10 (n.s.)			Δ 22 vs. 10 *	Δ 0 vs. −30		Δ 44.9 vs. 27.5 *
QoL	**MLHFQ** **60 vs. 81 ***	MLHFQ median −15.5 vs. −5 *	MLHFQ 24 ± 22 vs. 47 ± 22 *	SF 36 physical score 48.5 vs. 44.6 *	MLHFQ Δ 11 vs. 6 *		AFEQT 2.5 vs. 7.4 *	MLHFQ −5.4 *
Clinical outcome					Death: 8% vs. 18% *	**Death/hospitalization:** **28.5% vs. 44.6% ***	**Death/stroke/bleeding:** **9.0% vs. 12.3% ***	**Death/HF event:** **23.4% vs. 32.5%**

## Data Availability

No original data are presented in the current article.

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
