# Peer review of "Catheter Ablation for Atrial Fibrillation in Structural Heart Disease: A Review"

_jcm, 2023, doi:10.3390/jcm12041431_

Round 1

Reviewer 1 Report

Brasca et al. reported a review on AF ablation in structural heart disease.

The paper is well written and clear. No major language editing is needed.

However, I will extend the discussion on the CABANA subgroup with all the implications coming out from this study. 

I will also add a paragraph on the ablation strategies (PVI only vs PVI+). Are there evidences suggesting a more extensive approach?

Author Response

We thank the reviewer for its appreciation of our article and for its precious comments.

A more thorough discussion on the CABANA subgroup study has been added and moved to section 3.2 (lines 239-256)

Section 3.5 has been modified to accommodate additional considerations on CA techniques in SHD, including lesions beyond PVI.

Reviewer 2 Report

The authors gave a very comprehensive review of studies that analyzed catheter ablation in structural cardiomyopathies.

The manuscript is well written. The topic is relevant in this field. The conclusions are consistent and in concordance with the current clinical guidelines.

I have one additional comment/question. In the section CA in HFrEF authors have shown the most important studies that analyze this group of patients and CA was associated with improvement of LV function and quaility of life.

What about patients with coronary artery disease (CAD)?

One mentioned study (CAMERA-MRI) excluded patients with CAD.

Did any study include patients with ishaemic cardimyopathy and what were the results in this subgroup of patients?

Author Response

We thank the reviewer for its appreciation of our paper and for its generous comments.

Of all the randomized trials on AF and SHD, presented in Table 1, only the CAMERA-MRI excluded patients with CAD. On the other hand, this population was variably represented in the other trials (with a prevalence ranging from 23% to 73%) but, unfortunately, specific data on safety and efficacy in this subpopulation were not provided by the authors.

The only available evidence on the relationship between CAD and AF ablation is provided by single center retrospective studies that suggest an increased risk of AF recurrence and thromboembolic events following ablation in this population (10.1111/pace.14571 and 10.1111/jce.14029).

Although we acknowledge the relevance of this association and thank the reviewer for its precious remark, because of the scarce evidence available and the fact the CAD per se is not consistently considered a Structural Heart Disease (10.1093/ejcts/ezw313), which is the scope of the review, the authors would prefer not to include this topic in paper.